# Morpho-Molecular Features and Phylogenetic Relationships of *Metorchis butoridi* Oschmarin, 1963 (Trematoda: Opisthorchiidae) from East Asia

**DOI:** 10.3390/ani14010124

**Published:** 2023-12-29

**Authors:** Daria Andreevna Solodovnik, Yulia Viktorovna Tatonova, Vladimir Vladimirovich Besprozvannykh

**Affiliations:** 1Federal Scientific Center of the East Asia Terrestrial Biodiversity, Far Eastern Branch of the Russian Academy of Sciences, Pr-t 100-Letiya Vladivostoka 159a, 690022 Vladivostok, Russia; ytatonova@gmail.com (Y.V.T.); besproz@biosoil.ru (V.V.B.); 2Institute of Life Sciences and Biomedicine, FEFU Campus, Far Eastern Federal University, 10 Ajax Bay, Russky Island, 690922 Vladivostok, Russia

**Keywords:** *Metorchis* spp., Opisthorchiidae, morphology, phylogeny, 28S rRNA gene, ITS region, *cox*1 mtDNA gene

## Abstract

**Simple Summary:**

Morphological and molecular data were collected from experimentally reared adult trematodes of *Metorchis butoridi*, a parasite of the gallbladder of fish-eating birds from the Russian Far East. Phylogenetic relationships within the family Opisthorchiidae were analyzed. The results confirmed that the trematodes under study belong to the genus *Metorchis*. Some taxonomical issues of the genus are discussed.

**Abstract:**

Adult trematodes of the genus *Metorchis* were found in the gallbladders of ducklings that had been experimentally fed freshwater fishes of the genera *Rhynchocypris* and *Rhodeus* that were naturally infected by *Metorchis* metacercariae. Some of the trematodes were identified as *Metorchis ussuriensis*, whose morphology of developmental stages and molecular data had previously been described in detail. The other trematodes were confirmed as species *Metorchis butoridi* on the basis of morphological features: subterminal oral sucker, vitelline follicles with interrupted bands, and rosette-shaped testes. An analysis of phylogenetic relationships within Opisthorchiidae using nuclear and mitochondrial markers confirmed that the obtained trematodes were actually from the genus *Metorchis*. The morphological and molecular features indicated that a number of trematodes found in East Asia and described as *Metorchis orientalis* belong to *M. butoridi*. Also, the *M. orientalis* individuals from Europe are, in fact, representatives of another *Metorchis* species.

## 1. Introduction

Flukes of the family Opisthorchiidae are distributed throughout the world and are causative agents of severe infections. The features of invasion and pathogenesis of some opisthorchiids species, such as *Clonorchis* and *Opisthorchis* spp., have been sufficiently studied, but the epidemiology, taxonomy, and pathology of infections caused by trematodes of the genus *Metorchis* are still poorly examined [1,2]. The genus *Metorchis* Looss, 1899 comprises species that parasitize the liver’s bile ducts and the gallbladder in birds and mammals including humans [3]. This genus is represented by both conditional endemics and conditional cosmopolitans. This is explained by the fact that the differentiation of most *Metorchis* species is based mainly on morphometric data, without molecular evidence provided to confirm the taxonomic status. As a result, morphologically similar trematode individuals from the same or different localities can be attributed to both the same and different species. This viewpoint is supported by Sitko et al. [1], who showed, using a set of morphological and molecular data, that the flukes previously considered *Metorchis albidus* (Braun, 1893) and *Metorchis crassiusculus* (Rudolphi, 1809) are synonymous to *Metorchis bilis* (Braun, 1790). The results of Sitko et al. [1] and other studies indicate the importance of comprehensive research for the species identification of *Metorchis* spp. in local populations. This problem is also relevant for the East Asia region, where eight *Metorchis* species are listed: *Metorchis orientalis* Tanabe, 1921; *Metorchis taiwanensis* Morishita, 1929; *Metorchis elegans* Belogurov, Leonov, 1963; *Metorchis butoridi* Oschmarin, 1963; *Metorchis kimbangensis* Ha, 2005 (whose definitive host is the Amur catfish *Parasilurus asotus* (Linnaeus, 1758)); *Metorchis ussuriensis* Besprozvannykh, Tatonova, Shumenko, 2018; *Metorchis xanthosomus* Creplin, 1846; and *Metorchis pinquinicola* Skrjabin, 1913, of which the former six are Asian and the latter two are, respectively, European and African representatives [4,5,6,7,8,9]. The life cycles and morphology of developmental stages were described for both species native to East Asia, *M. taiwanensis* and *M. ussuriensis*. Molecular characteristics were only obtained from *M. ussuriensis* and *M. orientalis* [9,10,11,12,13,14,15].

Various authors have published data on the nuclear and mitochondrial genes of *M. orientalis*, along with the morphometric characteristics of adult individuals. The first data on molecular markers for the East Asian *Metorchis* species were provided by Ai et al. [11]. These authors considered the molecular markers of metacercariae from naturally infected fish in China to belong to *M. orientalis*. Subsequently, molecular data for flukes identified as *M. orientalis*, obtained using metacercariae or adult worms from naturally infected animals, have been published by Na et al. [12], Qiu et al. [13], Qiu et al. [14], and Wang et al. [15]. Except for the article by Wang et al. [15], other authors did not provide morphological data on the trematodes from which they had described molecular characteristics. Furthermore, the GenBank database contains a number of unpublished data for *M. orientalis* metacercariae collected in China from naturally infected cyprinids. As mentioned above, the life cycle, developmental stages, and molecular data were studied in another East Asian species, *M. ussuriensis* [9]. For other Asian *Metorchis* species, there are no available molecular data to date.

While conducting a parasitological survey of freshwater fish in the south of the Russian Far East, we found metacercariae morphologically similar to those of Opisthorchiidae Looss, 1899. To determine the taxonomical status of these metacercariae, we set up several experiments to rear adult flukes. The results of our study are presented below.

## 2. Materials and Methods

### 2.1. Sample Collection and Preparation of Whole Mounts

To obtain adult flukes, we used the following freshwater fishes naturally infected by metacercariae: the lake minnow *Rhynchocypris percnurus mantschuricus* (Berg, 1907) and the Amur bitterling *Rhodeus sericeus sericeus* (Pallas, 1776) from Lake Magdykovoe (Primorsky Krai, Russia; 45°57′ N, 133°53′ E) and *Rhynchocypris percnurus* (Pallas, 1814) from an unnamed reservoir near Bezymyannoe Village (Amur Oblast, Russia; 49°45′ N, 129°11′ E). An autopsy of 20 fish from both localities revealed a 100% prevalence of infection by opisthorchiid metacercariae, with an intensity of infection of 20 to 150 individuals per host. These infected fish were then fed to ducklings and rats, which are potential definitive hosts of Opisthorchiidae. These two classes of hosts were used simultaneously because in fields it is difficult to reliably determine opisthorchiid metacercariae on the basis of external morphology. Three one-week-old ducklings of the mallard *Anas platyrhynchos* dom. Linnaeus, 1758 and three laboratory common rat *Rattus norvegicus* (Berkenhout, 1769) were used in the experiment. Prior to the experiment, the animals were fed plant foods. Two ducklings and two rats were each fed ten fish (five *R. percnurus mantschuricus* and five *R. sericeus sericeus*) from Lake Magdykovoe; one duckling and one rat were fed the same numbers of *R. percnurus* from the unnamed reservoir near Bezymyannoe Village. Before being fed to the ducklings and rats, the fish were cut. After, the laboratory animals were dissected and examined for the presence of adult trematodes at 8 days post-infection (dpi) in the experiment with fish from Lake Magdykovoe and at 9 and 28 dpi (for the duckling and the rat, respectively) in the experiment with fish from the Bezymyannoe Village reservoir, respectively. The intestines of all the rats were free of flukes. However, the rats in the experiment with fish from Lake Magdykovoe and the rat in the experiment with fish from the Bezymyannoe Village reservoir had 7 juvenile and 15 adult *Clonorchis sinensis* Looss, 1907, in their bile ducts. Representatives of Echinostomatidae Looss, 1899, were found in the intestines of both ducklings in the experiment with fish from Lake Magdykovoe; one of these ducklings also had 73 and the other had 176 adult opisthorchiids in the gallbladder. The duckling in the experiment with fish from the Bezymyannoe Village reservoir had only five adult opisthorchiids in the gallbladder, but no trematodes were found in the bile ducts. The subsequent microscopic examination of the trematodes from the gallbladder revealed individuals of two opisthorchiid species. Most of them belonged to *M. ussuriensis* previously described from Lake Magdykovoe [9], and only 13 individuals were morphologically different from this species. Eleven trematodes were obtained from the fish from Lake Magdykovoe and two from the fish from the unnamed reservoir near Bezymyannoe Village. The trematodes were slightly rinsed in distilled water and then killed in hot distilled water. Several individuals were fixed in 70% ethanol for preparation of whole-mounts, and the others in 96% ethanol for DNA extraction. For morphological analysis, five adult individuals obtained from metacercariae parasitizing fish from Lake Magdykovoe were used to make whole-mounts. The trematodes were stained with carmine alum, dehydrated in a graded ethanol series, cleared in xylene, and mounted in Canada balsam on a slide under a coverslip.

### 2.2. DNA Extraction, Amplification, and Sequencing

Genomic DNA was extracted from the laboratory-reared adult trematodes using the HotSHOT technique [16]. The partial 28S rRNA gene (*28S*), the complete ITS1-5.8S-ITS2 rDNA region (ITS1-5.8S-ITS2), and the complete *cox*1 mtDNA (*cox*1) genes were amplified using polymerase chain reaction (PCR) (the pairs of primers are presented in Table 1). In addition, phylogenetic analysis was conducted using the partial *cox*1 gene (the nucleotide sequences of this marker were obtained for *M. ussuriensis* (GenBank: OR095177) and *Erschoviorchis anuiensis* Tatonova, Besprozvannykh, Katugina et al., 2020 (GenBank: OR084097), previously analyzed by Besprozvannykh et al. [9] and Tatonova et al. [17], respectively). The reaction mixture (10 μL of total volume) contained 0.25 mM of each primer, 5 μL GoTaq^®^ Green Master Mix (Promega Corporation, Madison, WI, USA), and approximately 10 ng total DNA in water. PCRs were performed on a GeneAmp 9700 thermocycler (Applied Biosystems, Waltham, MA, USA) and consisted of initial denaturation (1 min, 94 °C) followed by 35 cycles: denaturation (15 s, 94 °C), annealing (30 s, 55 °C for nuclear marker and 51 and 62 °C for mitochondrial marker), synthesis (2 min, 72 °C), and an additional extension for 5 min at 72 °C. PCR products were sequenced on an ABI 3500 Genetic Analyzer (Applied Biosystems, Waltham, MA, USA). External and internal primers were used for sequencing.

### 2.3. Analysis of Genetic Data

The nucleotide sequences of *M. butoridi* were assembled manually and aligned using the Clustal W tool in MEGA version 5.03 [24]. Eight partial *28S* sequences, two complete ITS1-5.8S-ITS2 sequences, and eight complete *cox*1 sequences were deposited in the GenBank database under the following accession numbers: OR095627–OR095634 (*28S*); OR095658–OR095659 (ITS1-5.8S-ITS2); and OR088530–OR088537 (*cox*1).

For genetic markers, *p*-distances and nucleotide and amino acid substitutions were analyzed using MEGA 5.0 software. The level of haplotype (*Hd*) and nucleotide (*SD*) diversity was estimated using DnaSP version 6.11.01 [25].

Phylogenetic relationships within Opisthorchiidae were assessed using partial *28S* and *cox*1 data. The phylogenetic tree within the genus *Metorchis* was reconstructed using the complete ITS1-5.8S-ITS2 and partial *cox*1 sequences. These reconstructions were performed using the Bayesian inference (BI) algorithm in the MrBayes program, version 3.1.2 [26]. The optimal evolutionary models were selected in jModeltest version 2.1.10 [27]; the F81+I+G model was optimal for all markers and reconstructions. The BI analyses were performed using 400,000 generations of Markov chain Monte Carlo (MCMC). This number of generations was sufficient as the SD value was ˂0.01. The chain was sampled at every 100th generation. The first 25% of trees produced were discarded as burn-in to permit the reconstruction of consensus trees.

## 3. Results

### 3.1. Taxonomic Summary

*Metorchis butoridi* Oschmarin, 1963.

Syn. *Metametorchis butoridi* (Oshmarin, 1963) Filimonova in Sonin, 1986 [28].

Definitive host: *Anas platyrhynchos* dom. (experimentally).

Localization: gallbladder.

Intensity of infection: two to seven individuals.

Second intermediate host: *Rhynchocypris percnurus mantschuricus*, *Rhynchocypris lagowskii*, and *Rhodeus sericeus sericeus* (natural hosts).

Locality: Lake Magdykovoe, Bolshaya Ussurka River basin (tributary of the Ussuri River), Primorsky Krai, south of the Russian Far East, 45°57′ N, 133°53′ E; unnamed reservoir near Bezymyannoe Village, Kupriyaniha Levaya River basin (Amur River basin), Amur Oblast, south of the Russian Far East, 49°45′ N, 129°11′ E.

Materials deposited: materials were deposited in the parasitological collection of the Zoological Museum, FSCEATB FEB RAS, Vladivostok, Russia (20 November 2020; e-mail: petrova@biosoil.ru).

### 3.2. Morphological Description

Adult individuals. Five specimens (Figure 1, Table 2). Body elongated. Adult specimens with narrow anterior half and slightly broader posterior half of body. Mature young specimens have rounded anterior and posterior ends of body with uniform width all along body length. Tegument with fine spines. Oral sucker terminal, cup-shaped. Ventral sucker round, equal to or smaller than oral sucker, located at border between anterior and middle thirds of body or in anterior part of middle third of body. Prepharynx absent. Pharynx small, spherical. Esophagus absent. Intestinal bifurcation immediately posteriorly of pharynx. Cecal circumflex located on external side of testes or dorsally to testes and reaches posterior end of body where cecal branches arranged close to each other. Testes two, lobbed, tandem, adjacent to each other, near posterior extremity, either on median line of body or slightly diagonal. Adult specimens have irregular anterior testis consisting of 5–6 large rounded or irregularly shaped lobes. Posterior testis rosette-shaped, with 6–7 round, oval, or pyramidal lobes. Both testes of mature young specimens rosette-shaped. Seminal vesicle tubular, curved. Genital pore median, immediately anteriorly of ventral sucker. Ovary globular or transversally oval, pretesticular, or at level of anterior lobes of anterior testis, either left to median line or on median line. Seminal receptacle right of ovary; Mehlis’ gland left of ovary. Uterus extends from ovary to slightly anteriorly of genital pore. Uterine coils extend laterally, covering ceca. Vitellarium consists of two fields extending from anterior boundary of anterior testis, or from middle of anterior testis, or from level of ovary and reaching middle of forebody. Vitelline follicles small, elongate or transversally oval. Follicles in vitelline fields combined into 5–7 bands separated from each other. Eggs small, numerous, embryonated, operculated, with knob. Excretory vesicle Y-shaped.

### 3.3. Genetic Data

#### 3.3.1. Intraspecific Variation

The length of the partial sequence of the 28S rRNA gene for eight specimens of *M. butoridi* was 1049 bp. An analysis of these samples indicated a 99.9% identity and the presence of one G → A transition at position 353 bp for one individual from Amur Oblast.

Nucleotide sequences of the ITS1-5.8S-ITS2 rDNA region were obtained for two *M. butoridi* specimens. The total length of the region was 1076 bp, where ITS1, 5.8S, and ITS2 amounted to 629, 160, and 291 bp, respectively. ITS1-5.8S-ITS2 did not have nucleotide substitutions within species.

The total length of the *cox*1 mtDNA gene of *M. butoridi* was 1548 bp (5′-end was detected using a mitochondrion of *M. orientalis* KT239342; 3′-end was detected by the presence of a termination codon TAG). There were no recorded nucleotide differences between individuals from different localities, and each of the eight *M. butoridi* samples (six from Primorsky Krai and two from Amur Oblast) represented a unique haplotype (*Hd* = 1.000, *SD* = 0.063). The intraspecific genetic distances varied from 0.1 to 1.1% between the *M. butoridi* individuals. Among the eight sequences, 22 variable sites were found, of which 11 were only in the OR088531 sample and were absent from other individuals. The two individuals from Amur Oblast differed by 15 nucleotide substitutions, 5 of which led to amino acid substitutions. Among the six individuals from Primorsky Krai, 10 nucleotide substitutions were found, of which 2 led to amino acid substitutions. The largest number of nucleotide substitutions among the *M. butoridi* individuals was found between two individuals from Primorsky Krai (OR088534) and Amur Oblast (OR088531) (17 nucleotide and 7 amino acid substitutions).

#### 3.3.2. Phylogenetic Relationships

A phylogenetic analysis for *28S* data was performed using the sequences of Opisthorchiidae species published in the NCBI database. A representative of Heterophyidae Leiper, 1909 was used an outgroup (Figure 2). The total length of the analyzed *28S* partial sequence was 1072 bp. In the phylogenetic reconstruction, three species of the genera *Metorchis*, *M. orientalis*, *M. ussuriensis*, and *M. butoridi*, formed a high-supported branch with unresolved topology. The genetic distances between the *Metorchis* representatives varied within 0.3–0.6%, with the highest difference observed between *M. orientalis* and *M. ussuriensis* (Table 3). According to the genetic distances, *M. butoridi* was the closest to *M. orientalis*. The representatives of the genera *Clonorchis* Looss, 1907 and *Opisthorchis* Blanchard, 1895 formed a sister branch to *Metorchis*; the genetic distance between these branches was 1.3%. Within the *Clonorchis* and *Opisthorchis* branches, the maximum distance (1.7%) was detected between *Opisthorchis noverca* Braun, 1902 vs. *Opisthorchis felineus* (Rivolta, 1884) Blanchard, 1895 and *Opisthorchis viverrini* (Poirier, 1886), and the minimal distance was between *C. sinensis* and *O. felineus*, 0.6%.

The reconstruction of phylogenetic relationships based on the complete ITS1-5.8S-ITS2 rDNA region (1123 bp) divided the *Metorchis* representatives into three clusters with high statistical support (Figure 3). The complex Cluster 1 combined *M. butoridi* and flukes previously indicated as *M. orientalis* in various publications [11,13,14,15,30]. In this cluster, *M. orientalis* KX832894 from Heilongjiang Province of China formed an independent branch (Group IV) that differed from the other *Metorchis* representatives in Cluster 1 by 0.1–0.7%. The rest of the individuals in Cluster 1 constituted a single branch with low statistical support. In the structure of this branch, *M. butoridi* and the *M. orientalis* individuals MT231323, MK482055, MK482051, and MW828729 from Jilin and Heilongjiang Provinces did not make up a separate branch. These samples formed a brush-like structure in Cluster 1, and, therefore, they were tentatively designated as Group III. Samples in Group III did not have genetic differences in the ITS1-5.8S-ITS2 rDNA region. Group I in Cluster 1, being the most variable, included the specimens of *M. orientalis* from the southern Guangxi Province in China. Group I had a high level of statistical support and differed from the other groups in Cluster 1 by 0.5–0.8%. Group II was less variable and formed a highly supported branch, including three *M. orientalis* individuals from Heilongjiang Province (MK482052, KX857496, and MK482054). Group II was 0.3–0.8% genetically different from the other groups in Cluster 1.

Cluster 1 differed from Cluster 2 and Cluster 3 by 2.0 and 2.1%, respectively (Figure 3). Cluster 2 combined one *M. ussuriensis* individual and four specimens from Denmark referred to as *M. orientalis* by Duan et al. [31]. The genetic distances among the Danish flukes from Cluster 2 were within 0.0–0.2%. The minimum and maximum values between the Danish flukes and *M. ussuriensis* were 0.3 and 0.5%, respectively. Another specimen of *M. orientalis* from Denmark stood out as an independent branch (Cluster 3). This specimen was genetically different from the Danish *M. orientalis* from Cluster 2 by 1.5–1.7% and from *M. ussuriensis* from Cluster 2 by 1.9%; in general, the genetic distance between Cluster 2 and Cluster 3 was 1.6%.

For all the complete ITS1-5.8S-ITS2 rDNA region sequences included in the phylogenetic analysis, 50 variable sites were identified (Figure 4). There were no nucleotide substitutions within the genus *Metorchis* in 5.8S rRNA gene sequences, and the ITS1 and ITS2 regions amounted to 35 and 15 substitutions, respectively. Each cluster presented in the phylogenetic reconstruction (Figure 3) had unique nucleotide substitutions fixed in all members within the cluster. Cluster 1 had 10 unique substitutions, of which 6 were located in the ITS1 rDNA region. In addition, Group I and Group II of Cluster 1 had unique nucleotide substitutions observed in all individuals of the groups and not found in the other groups and clusters (Figure 4). Thus, for southern China individuals from Group I, 11 fixed substitutions and 1 bp indel at position 103 bp were found in the ITS1 rDNA region. Group II had only one unique substitution in ITS1, and no unique fixed substitutions were recorded from Groups III and IV. Cluster 2 had five unique nucleotide substitutions fixed in all individuals within it: at positions 632 and 633 bp of the ITS1 region and at positions 102, 104, and 106 bp of the ITS2 region. *Metorchis ussuriensis* differed from the Danish *M. orientalis* (Cluster 2) by three nucleotide substitutions (119 and 235 bp in ITS1 and 1112 bp in ITS2) and the indel CAG in the ITS2 region. Only one sample in Cluster 3 of *M. orientalis* had 12 unique nucleotide substitutions, of which 7 were present in ITS1; this sample also had a unique indel at position 1064–1066 bp of the ITS2 region.

We also performed a phylogenetic reconstruction for Opisthorchiidae based on the partial sequences of the *cox*1 mtDNA gene (1044 bp) (Figure 5). Members of the genera *Metorchis*, *Opisthorchis*, and *Clonorchis* formed a cluster with low statistical support. In this cluster, the lowest value of genetic difference (10.1%) was between *M. orientalis* and *O. viverrini* (Table 4), which formed a highly supported branch. The lowest value (13.6%) of genetic distance within the genus *Opisthorchis* was found between *O. felineus* and *Opisthorchis sudarikovi* Feizullaev, 1961. In the phylogenetic reconstruction, *M. butoridi* clustered with *M. ussuriensis*, with the genetic distance between these species being 15.0%. *Metorchis butoridi* genetically differed by 16.4% from *M. orientalis* and by 14.0–15.9% from *Opisthorchis* spp.

The NCBI database contains short sequences of the *cox*1 mtDNA gene for most representatives of the genus *Metorchis*. Therefore, phylogenetic relationships within the genus were analyzed based on partial 301 bp sequences. In the phylogenetic reconstruction, the *Metorchis* representatives formed five clades (Figure 6). As in the case of the reconstruction of relationships within the Opisthorchiidae, *M. orientalis* KT239342 from China was classified as a separate Clade 1 and it occupied a basal position relative to the other *Metorchis* specimens. In Clade 5, the individuals of *M. orientalis* from the northern and southern provinces of China were combined with those of *M. butoridi*. The genetic distances between specimens in this clade varied within 0.0–3.3%. Other members of the genus *Metorchis* included in the analysis were Clade 2 with *M. ussuriensis*, Clade 3 with *M. xanthosomus*, and Clade 4 that comprised the sequences of both *M. bilis* and *M. albidus*. In Clade 4, the samples were split into two branches, with the genetic distance between them being 2.6% and no amino acid substitutions. The positions of nucleotide and amino acid substitutions were also determined for the partial *cox*1 sequences of *M. orientalis* and *M. butoridi* included in the phylogenetic analysis (Figure 7). A total of 50 nucleotide substitutions were found. *Metorchis orientalis* KT23934238 had 38 nucleotide substitutions out of 50, with 7 of them being nonsynonymous and resulting in 6 amino acid substitutions. All the *M. orientalis* individuals from southern Guangxi Province of China (HM347229–HM347234) had ten unique fixed nucleotide substitutions, of which seven resulted in six amino acid substitutions. Two fixed nucleotide substitutions in *M. orientalis* KY232050–KY232053 and KY232056, were represented by transitions and did not affect the amino-acid composition.

## 4. Discussion

Based on their morphological characteristics, the experimentally reared adult trematodes matched the diagnosis of the genus *Metorchis*. Among representatives of this genus, the individuals in our material showed the greatest morphological similarity with East Asian species such as *M. orientalis*, *M. taiwanensis*, *M. elegans*, and *M. butoridi*. Available data for adult *Metorchis* flukes indicate, on the one hand, their significant morphological similarity and, on the other, their characteristic variability in the length of vitelline fields, the shape of testes, the length of the uterus, and metric data [32,33]. In the latter case, this was also evidenced by significant differences between individuals attributed to the same species, which are presented in Table 2. The trematodes in our study, similar to *M. taiwanensis*, had elongated body, an almost uniform width all along their length, and no hiatus between individuals in terms of metric characteristics. However, they could not belong to *M. taiwanensis*, since they differed from this species in the position of the oral sucker (terminal vs. subterminal in *M. taiwanensis*) and in the structure of the vitellarium (vitelline follicles forming 5–7 interrupted bands vs. vitelline follicles merged into continuous lateral fields in *M. taiwanensis*). Another species, *M. elegans*, in contrast to the above-listed trematodes, has a massive pharynx equal in size to the oral sucker (Table 2), which did not allow us to attribute the individuals in our material to this species. The trematodes in the present study also had significant differences from individuals of *M. orientalis* from the type locality in Japan [34]. The differences between them were found in the shape of the body (elegantly elongated vs. wide, tongue-shaped in *M. orientalis*), in the esophagus (absent vs. present, respectively), in the structure of the testes (deeply lobed, consisting of rounded, oval, pyramidal lobes forming a rosette vs. irregularly shaped, lobed, respectively), in the location of the oral sucker (terminal vs. subterminal, respectively), and in the vitellarium structure (vitelline follicles forming 5–7 interrupted bands vs. vitelline follicles merged into continuous lateral fields, respectively). The most extensive information about flukes identified as *M. orientalis* was obtained from China and the Korean Peninsula. The first record of *M. orientalis* from China dates back to 1938 [35]. Unfortunately, that publication was not available to us. However, subsequently, when Chen and Tang [7] examined ducks for natural infection by representatives of *Metorchis* in China, they found two species in the birds’ gallbladders, *M. orientalis* and *M. taiwanensis*, for which they provided morphometric data. The trematodes that Chen and Tang identified as *M. orientalis* were, indeed, similar to Japanese individuals of this species in terms of the shape of the body, the body length-to-width ratio, the presence of a subterminal sucker and the esophagus, the lobed testes, and the continuous vitelline fields. In the Korean Peninsula, flukes identified as *M. orientalis* were reared experimentally by feeding chickens with fish naturally infected by metacercariae [29]. With this method for obtaining adult worms, there is a high probability of a mixed infection of fish by metacercariae of different opisthorchiids, which results in several species of *Metorchis* simultaneously parasitizing the definitive hosts. The probability of mixed infection is evidenced by the data obtained by Chen and Tang [7] and in our study. Furthermore, the flukes in the drawing by Sohn et al. [29] have a number of morphological differences in the length and structure of the vitelline fields and in the testes structure, which gives reason to assume that the individuals drawn belong to different species. Wang et al. [15] also published data on the morphology of adult flukes found in a black swan from China and referred to them as *M. orientalis*. By their morphological characteristics, those individuals had a number of significant differences from *M. orientalis* in the oral sucker position and in the structure of the testes and vitellarium. However, they were identical in these characteristics to the trematodes in our material: the sucker located terminally; the testes rosette-shaped with oval or pyramidal lobes; and the vitelline follicles forming interrupted bands. These features were clearly distinguishable in the photograph and drawing of trematodes presented by Wang et al. [15]. On the basis of the above-indicated morphological characteristics, the individuals in our material and those obtained by Wang et al. [15] corresponded to the *M. butoridi* specimens described by Oschmarin [6] from the striated heron *Butorides striata* (Linnaeus, 1758) inhabiting the south of the Russian Far East. As for *M. butoridi*, in addition to the Far East of Russia and China, there are documented records of this species from the great cormorant *Phalacrocorax carbo* (Linnaeus, 1758) and the gull *Larus* spp. Linnaeus, 1758, from Eastern Siberia [28,36,37]. However, no morphological and molecular data for these trematodes are available to date.

It is also worth noting that in 1986, *Metorchis butoridi* was assigned to the genus *Metametorchis* Morozov, 1939 [28], which was subsequently recognized as a synonym of *Parametorchis* Price, 1929 [3]. However, the trematodes described by Oshmarin and obtained in the present study belong to the genus *Metorchis* because their uterine loops do not form a rosette and the vitelline fields reach the level of the ovary or anterior testes in contrast to those in *Parametorchis*. Moreover, the belonging of trematodes in the present study to the genus *Metorchis* has also been confirmed using molecular data.

For the adult *Metorchis* trematodes that we found in the south of the Russian Far East, we obtained molecular characteristics for nuclear and mitochondrial markers along with morphological data. Phylogenetic relationships for the species of the genus *Metorchis* and the family Opisthorchiidae were assessed on the basis of nucleotide sequences.

It is commonly known that, in some cases, *28S* can be used as an efficient marker for differentiating trematode species, while in others, its efficiency becomes evident only at intergeneric or higher taxonomic levels [38,39,40]. With the use of this marker, the differences between *M. orientalis*, *M. ussuriensis*, and the individuals in our material identified as *M. butoridi* were 0.3–0.6% (Figure 2). In our earlier studies that took into account the data for *Cryptocotyle* Lühe, 1899, previously included in Opisthorchiidae, interspecific distances within this family were in a range of 1.6–6.1% [17]. However, it should be noted that this level of differentiation was determined by the lack of molecular data on *28S* for species of the genus *Metorchis*. Nevertheless, in our present study, representatives of different genera, *C. sinensis* and *O. felineus*, differed only by 0.6%, although a long sequence of the 28S rRNA gene was analyzed. In this case, if we assume that *C. sinensis* and *O. felineus* belong to different genera, then the differences in the nuclear marker between *M. orientalis*, *M. ussuriensis*, and *M. butoridi* correspond at least to the interspecific level for Opisthorchiidae. The low sensitivity of the 28S rRNA gene for species differentiation was also evidenced by the values of interspecific differences for species of *Metagonimus* Katsurada, 1913, from the sister family Heterophyidae, some of which did not differ in this marker [41]. Thus, in the present study, the 28S rRNA gene showed the interspecific level of differences between the *Metorchis* species. However, only three of them were included in the analysis. To confirm these results, we also analyzed more sensitive markers.

As for the nuclear markers ITS1 and ITS2, their significance for the purpose of differentiation of Opisthorchiidae species is also ambiguous. Kang et al. [42] considered ITS1 as a powerful marker for distinguishing between these species. In the publication by Besprozvannykh et al. [9], the genetic distances in this marker for *M. ussuriensis* were 0.4 and 0.6% from, respectively, *M. bilis* and *M. xanthosomus*, which fits into the range of genetic intraspecific distances in opisthorchiids. Xiao et al. [43] showed that the intraspecific variation in *C. sinensis* for the ITS1 marker reached 1.7%. With the use of the ITS2 marker, the intraspecific variation in *O. viverrini* was from 0.9 to 1.8% [21], while the genetic distance between *Opisthorchis lobatus* (Bilqees, Shabbir, Parveen, 2003) and *O. viverrini* was 0.9% [44]. However, it should be noted that, when estimating intraspecific distances for *O. viverrini*, Katokhin et al. [21] questioned the species identity of some of the trematode specimens. On the other hand, the second spacer region (ITS2) did not differentiate *M. ussuriensis* and *M. bilis* at all [9]. Thus, according to different authors, interspecific distances of the family Opisthorchiidae based on the ITS1 and ITS2 markers vary within a wide range, and, therefore, much more information on these regions of ribosomal cluster is required to determine the efficiency of each of these markers in differentiating certain groups of trematodes within this family.

The results of our analysis of phylogenetic relationships for the complete ITS1-5.8S-ITS2 rDNA region among the *Metorchis* species showed that *M. butoridi* and the flukes identified as *M. orientalis* from the northern provinces of China (Jilin and Heilongjiang), including the adult individuals of *M. orientalis* extracted from a black swan [15] (Figure 3: Cluster 1: Group III), which had identical nucleotide sequences except for a single C → T transition at position 53 bp in *M. orientalis* MK482053 (Figure 3; Figure 4). Judging by the morphological identity of the adult trematodes in our material, the individuals from the black swan in China, and the *M. butoridi* flukes described by Oschmarin [6], as well as by the lack of molecular differences between the former two, all specimens in Group III of Cluster 1 belonged to the same species, *M. butoridi*. Unfortunately, for the rest of the individuals identified as *M. orientalis*, included in Group III of Cluster 1, no morphological characteristics were provided [14,30]. The representative of the genus from Group IV of Cluster 1 most likely also belonged to the species *M. butoridi* (with no morphological data available either) [13], since its differences from the specimens of Group III of Cluster 1 were insignificant and amounted to only 0.1%. It is worth noting that all the specimens in Group III and Group IV of Cluster 1 were collected from rivers of the Amur River basin in China and Russia, whose middle and lower reaches are, apparently, the distribution range of *M. butoridi*.

The remaining sequences of the ITS1-5.8S-ITS2 rDNA region from the individuals collected in the northern and southern provinces of China, identified as *M. orientalis* (Figure 3: Cluster 1: Group I and Group II), were obtained using both mature individuals and metacercariae from naturally infected fish, and no morphological data were provided for them [13,14,31]. The specimens of Group I and Group II differed from those of Group III of Cluster 1 by 0.5 and 0.3%, respectively, and from each other, by 0.8%. Taking into account the previously obtained data for representatives of *Metorchis* [9], such values of genetic differences may indicate that the individuals from Heilongjiang and Guangxi Provinces, combined into Group I and Group II of Cluster 1, were a distinct species (Figure 3). The species independence of the specimens in Group I and Group II was also confirmed by the presence of fixed nucleotide substitutions and indels (Figure 4). However, reliable species identification of these individuals will only be possible with the availability of both morphological data for adult individuals from the abovementioned provinces of China and molecular data that would match those obtained for metacercariae.

The issue of the species affiliation of the flukes found in Denmark and attributed to *M. orientalis* is also rather challenging [31]. Molecular data were obtained using cercariae from naturally infected mollusks collected at two remote localities in Denmark. In the phylogenetic reconstruction based on the ITS1-5.8S-ITS2 rDNA region, the difference between the specimens of Cluster 2 and Cluster 3 (Figure 3) reached 1.5%, and the individuals from each cluster had unique fixed substitutions (Figure 4), which might indicate their different species affiliations. However, their validity could not be confirmed, as the genetic data were derived from cercariae. Undoubtedly, the flukes found in Denmark belonged to the genus *Metorchis*. However, with the lack of morphological data for mature individuals and according to the analysis of the phylogenetic reconstruction, these could not be attributed to *M. orientalis*.

In addition to the Danish flukes, Cluster 2 (Figure 3) also included *M. ussuriensis*, which differed by 0.3% from them. However, unlike the specimens from Denmark, *M. ussuriensis* had a unique feature: the CAG indel fixed in the ITS2 sequences of all specimens of the species studied so far (Figure 4). This fact indicates that these individuals belong to different species. In addition, the authors’ indication of gastropods of the family Lymnaeidae [31] as the first intermediate hosts of the Danish *Metorchis* specimens is in poor agreement with the reports that Bithyniidae snails play this role in the life cycle of *M. ussuriensis*, which also confirms the independence of *M. ussuriensis*.

Molecular data for both the long and short regions of the mitochondrial marker *cox*1 also supported the validity of *M. butoridi*, as was previously inferred from analysis of nuclear DNA sequences. In the phylogenetic reconstruction based on the long sequence of the *cox*1 gene, the trematodes obtained in this study formed a single cluster with *M. ussuriensis*, from which they differed by 15% (Figure 5). Of greater interest is the reconstruction based on the short sequences of the mitochondrial gene, as it included more *Metorchis* species (Figure 6). The distances between five clades in this reconstruction varied from 7.6 to 15.4%, while the distances within clades were no greater than 1.5%. One of the *cox*1 sequences of an adult individual from China identified as *M. orientalis* (KT239342, Clade 1) occupied a basal position relative to the other representatives of *Metorchis*. However, in addition to the long *cox*1 sequence, this specimen formed a single cluster with *O. viverrini* (Figure 5) but not with *M. butoridi* or *M. ussuriensis*. The lack of morphological characteristics for this trematode did not allow for a comparative analysis, neither with our samples of *M. butoridi* nor with other members of the genus. The remaining *M. orientalis* sequences from China appeared together with *M. butoridi* in Clade 5 (Figure 6). It should be noted that all mitochondrial data for *M. orientalis* in this cluster were obtained using metacercariae from naturally infected fish caught in the northern (Heilongjiang) and southern (Guangxi) provinces of China. Due to the lack of information on the morphology of adult individuals, their taxonomic status was based only on molecular data. In Clade 5 in Figure 6, among the *M. orientalis* specimens from Heilongjiang Province, there are both individuals identical to *M. butoridi* in terms of genetic distance (Haplotype 1) and individuals that differ from *M. butoridi* by 0.7% (Haplotype 2) (Figure 6). This difference value corresponds to the intraspecific level and is at least 10-fold lower than the differences between the *Metorchis* species. Furthermore, no nonsynonymous substitutions were found between these two haplotypes, which also indicates that they belong to the same species. In Clade 5 (Figure 6), a branch with a high support was also formed with the specimens of *M. orientalis* from Guangxi (HM347229–HM347234) that differed from the other individuals in the clade by 10 nucleotides and six amino acid substitutions. This number of amino acid substitutions corresponds to the interspecific level of differences within the family Opisthorchiidae [9]. However, the genetic distance between *M. orientalis* from Guangxi and the other *Metorchis* specimens in Clade 5 was no greater than 4.1%, i.e., significantly lower than the minimum interspecific level among other species of this genus. Despite the low genetic distances, the large number of fixed nucleotide and amino acid substitutions in the *M. orientalis* specimens from Guangxi may indicate that they belong to a separate species. The question of their species status remains open, and therefore, data are needed on the morphology of adult trematodes that would be genetically similar to the individuals from Guangxi. Further molecular studies of *Metorchis* species should provide a wider coverage, which may clarify interspecific differences within this genus.

## 5. Conclusions

According to our findings, the adult trematodes that were reared experimentally and completed their life cycle in the southern Russian Far East resemble the Far Eastern species *M. butoridi* morphologically. An analysis of morphological and genetic data revealed that a significant number of nucleotide sequences from the GenBank database that were previously identified as *M. orientalis* are *M. butoridi* and not as indicated. The current study therefore underscores the importance of incorporating both morphological and genetic information—a comprehensive approach—in making taxonomic classifications of trematodes. Thus, it might be challenging to identify a wide range of species, including most trematode taxa, considering that they are especially pertinent to an integrated identification strategy.

## Figures and Tables

**Figure 1 animals-14-00124-f001:**
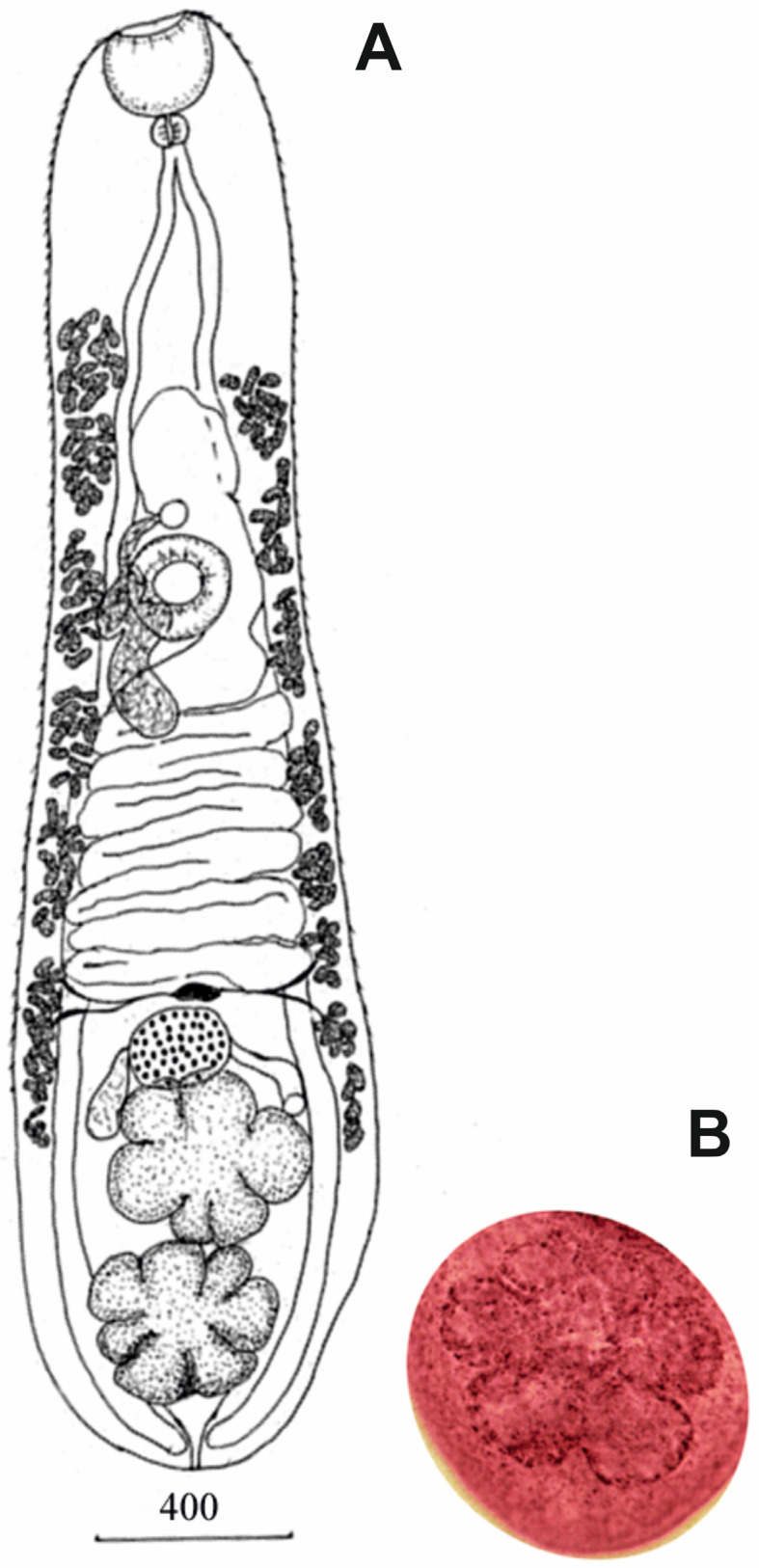
(**A**) Adult *Metorchis butoridi*, scale bar = 400 μm; (**B**) posterior rosette-shaped testis (whole-mount).

**Figure 2 animals-14-00124-f002:**
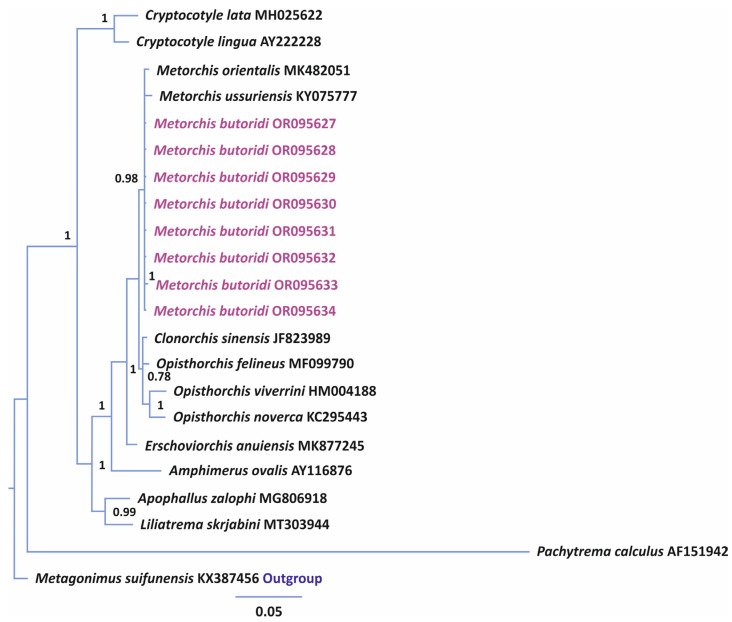
Phylogeny of the family Opisthorchiidae reconstructed using the 28S rRNA gene sequences with the Bayesian inference (BI) method. Bayesian posterior probabilities of ≥0.50 are shown. *Metorchis butoridi* samples obtained in this study are marked in purple, the outgroup is in blue.

**Figure 3 animals-14-00124-f003:**
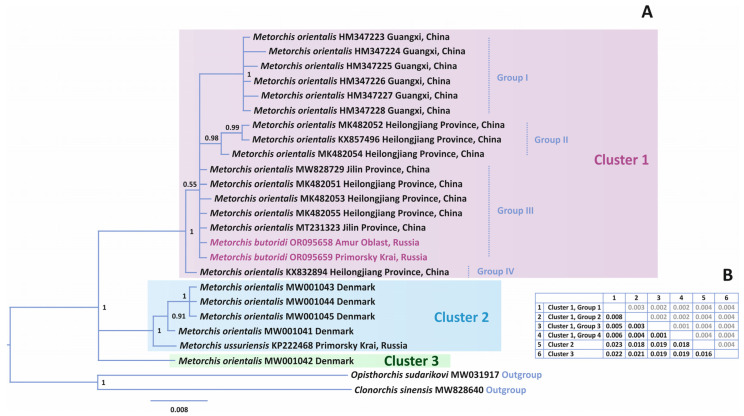
(**A**) Phylogeny of the genus *Metorchis* reconstructed using complete ITS1-5.8S-ITS2 rDNA region sequences with the Bayesian inference (BI) method. Bayesian posterior probabilities of ≥0.50 are shown. *Metorchis butoridi* samples obtained in this study are marked in pink, outgroup samples are in blue. Three clusters are labelled by different colors. (**B**) Genetic *p*-distances (below the diagonal; black) and standard error estimates (above the diagonal; gray) between clusters and groups.

**Figure 4 animals-14-00124-f004:**
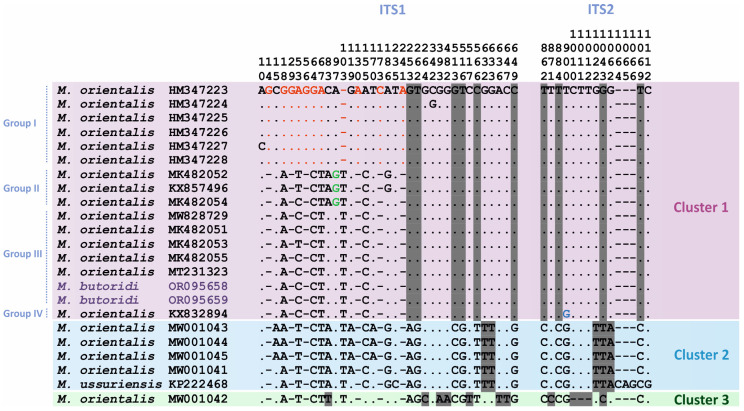
Variable sites of nucleotide sequences of the complete ITS1-5.8S-ITS2 rDNA region for the *Metorchis* species. The clusters and clades correspond to those in Figure 3. Three clusters are labelled by different colors. Gray rectangles show unique fixed substitutions inherent in the clusters. Red, green, and blue colors indicate unique nucleotide substitutions in groups within Cluster 1. *Metorchis butoridi* samples are marked in purple. Dots and hyphens indicate invariable sites, and indels, respectively. The 5.8S rRNA gene is not marked in the figure because it does not contain nucleotide substitutions between all the species.

**Figure 5 animals-14-00124-f005:**
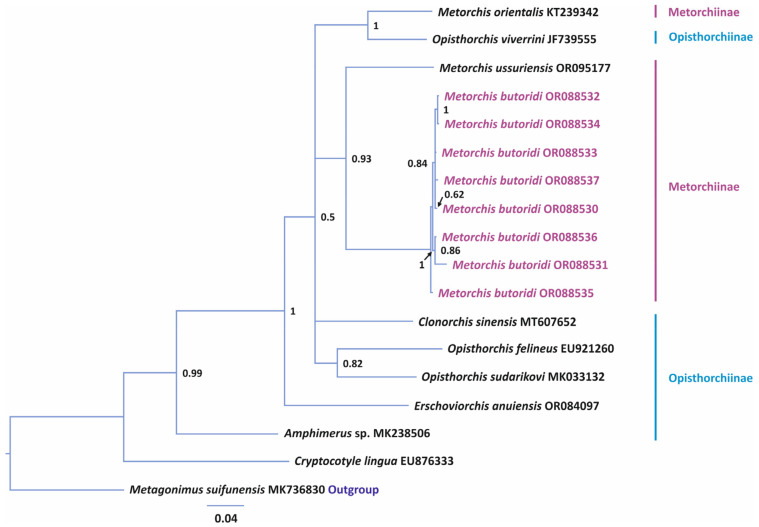
Phylogeny of the family Opisthorchiidae reconstructed using partial *cox*1 gene sequences with Bayesian inference (BI) method. Bayesian posterior probabilities of ≥0.50 are shown. *Metorchis butoridi* samples obtained in this study are marked in purple, the outgroup is in blue.

**Figure 6 animals-14-00124-f006:**
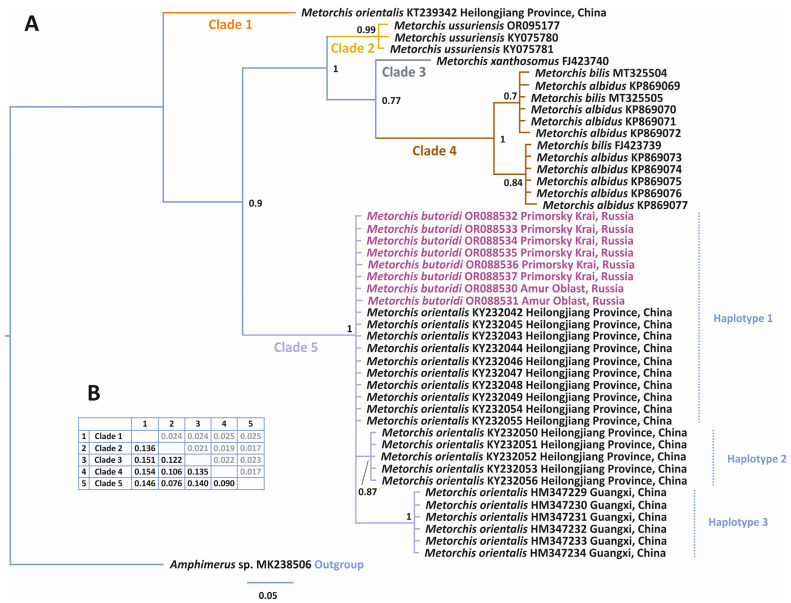
(**A**) Phylogeny of the genus *Metorchis* reconstructed using partial *cox*1 gene sequences with the Bayesian inference (BI) method. Bayesian posterior probabilities of ≥0.50 are shown. *Metorchis butoridi* samples obtained in this study are marked in purple; the outgroup is in blue (**B**) Genetic *p*-distances (below the diagonal; black) and standard error estimates (above the diagonal; gray) between clades.

**Figure 7 animals-14-00124-f007:**
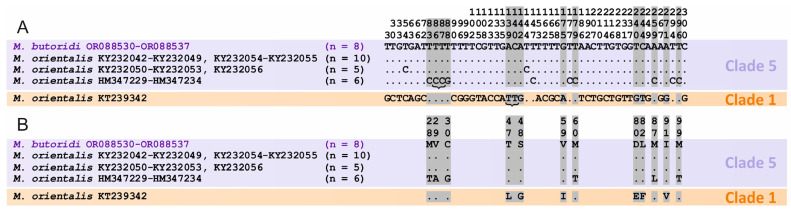
Variable sites of nucleotide (**A**) and amino acid (**B**) sequences of the partial *cox*1 gene for the *Metorchis* species. Colors indicate clades corresponding to those in Figure 6; gray rectangles show nonsynonymous substitutions; *n*—number of samples. *Metorchis butoridi* samples are marked in purple. Dots indicate invariable sites, curly brackets indicate localization of two nucleotide substitutions which lead to one amino acid substitution.

**Table 1 animals-14-00124-t001:** The list of primers used for PCR and sequencing reactions. Asterisk indicates primers reconstructed by the authors using the IDT service (OligoAnalyzer Tool: https://eu.idtdna.com/calc/analyzer (accessed on 7 July 2023)).

Gene	28S	ITS1-5.8S-ITS2	*cox*1
**PCR**	forward dig12 (5′-AAG-CAT-ATC-ACT-AAG-CGG-3′) and reverse 1500R (5′-GCT-ATC-CTG-AGG-GAA-ACT-TCG-3′) [18]	forward BD1 (5′-GTC-GTA-ACA-AGG-TTT-CCG-TA-3′) [19] and reverse 28S4R (5′-TAT-TTA-GCC-TTG-GAT-GGA-GTT-TAC-C-3′) [9]	forward 5cF22 (5′-TAG-ACT-ATC-TGT-CTT-CAA-AAC-A-3′) [20] and reverse CO1-Rv (5′-AAC-AAA-TCA-TGATGC-AAA-AGG-TA-3′) [21]; forward MBCOIF* (5′-GTG-TCT-GCG-GTA-TGG-CTC-CT-3′) and reverse MBCOIR* (5′-AGC-ACC-CAG-TCC-ACA-CAA-GG-3′)
**Sequencing** **(internal)**	900F (5′-CCG-TCT-TGA-AAC-ACG-GAC-CAA-G-3′), ECD2 (5′-CTT-GGT-CCG-TGT-TTC-AAG-ACG-GG-3′) [18]; 1200R (5′-GCA-TAG-TTC-ACC-ATC-TTTCGG-3′) [22]; 28S4R	F1i27* (5′-TGC-CTA-CCC-GTC-TGA-TGC-TCT-C-3′); R1i15 (5′-GTC-GTA-ACA-AGG-TTT-CCG-TA-3′) [23]; F2i03* (5′-GAC-TGC-CTA-GAT-GAG-GGG-GTG-G-3′); R2i18* (5′-GAC-AAA-GGA-CCA-ACA-ACG-GAG-C-3′)	MOR11.1* (5′-GAG-TTC-CAG-CAA-GCA-TAA-AC-3′)

**Table 2 animals-14-00124-t002:** Measurements for adult flukes of the genus *Metorchis* (in mm; min–max range); *n*—number of specimens. This table is based on Table 2 in Besprozvannykh et al. [9].

	*M. butoridi* Our Date *n* = 5	*M. butoridi* [6]	*M. orientalis* [4]	*M. orientalis* [7]	*M. orientalis* [29]	*M. orientalis* [15]	*M. ussuriensis* [9]	*M. elegans* [5]	*M. taiwanensis* [7]	*M. taiwanensis* [10]
**Body length**	1.83–3.00	4.10–4.70	2.36–4.65	3.00–6.81	3.43	4.98–9.10	2.42–3.93	3.20–4.50	3.49–5.81	2.50–3.50
**Body width**	0.35–0.60	0.65–0.70	0.53–1.23	0.61–1.64	0.83	0.61–1.35	0.34–0.79	0.46–0.54	0.42–1.45	0.40–0.55
**Length/width ratio, %**	1:4.06–6.32	–	–	1:2.99–4.49	–	–	1:4.06–8.50	–	1:4.26–11.2	–
**Forebody length**	0.63–1.08	–	–	0.66–1.81	–	–	0.83–1.42	–	0.83–1.68	–
**Body/forebody length ratio, %**	1:2.70–3.56	–	–	–	–	–	1:2.30–3.20	–	–	–
**Oral sucker length**	0.13–0.19	0.22–0.30	0.18–0.29	0.27–0.38	0.28	0.33–0.43	0.16–0.19	0.25–0.25	0.16–0.29	0.18–023
**Oral sucker width**	0.22–0.25	0.30–0.34	0.18–0.29	0.30–0.38	0.29	0.31–0.37	0.17–0.19	0.25–0.32	0.20–0.28	0.20–0.23
**Ventral sucker length**	0.16–0.19	0.25–0.34	0.12–0.30	0.23–0.33	0.27	0.30–0.40	0.14–0.18	0.21–0.27	0.20–0.28	0.15–0.18
**Ventral sucker width**	0.16–0.20	0.25–0.34	0.12–0.30	0.22–0.33	0.27	0.27–0.39	0.14–0.17	0.19–0.25	0.20–0.27	0.15–0.18
**Ventral/oral sucker length ratio, %**	1:1.00–1.39	–	–	–	–	–	1:0.82–0.98	–	–	–
**Ventral/oral sucker width ratio, %**	1:0.70–0.86	–	–	–	–	–	1:0.78–0.94	–	–	–
**Pharynx length**	0.05–0.07	0.08–0.10	0.04–0.06	0.07–0.08	0.07	–	0.05–0.07	0.30	0.07–0.10	–
**Pharynx width**	0.05–0.07	0.08–0.09	0.04–0.06	0.07–0.08	0.07	–	0.05–0.07	0.26	0.05–0.08	–
**Esophagus length**	0	0	–	0.08–0.18	0.07	–	0.03–0.10	–	0.03–0.12	–
**Ovary length**	0.10–0.17	0.18–0.20	0.26–0.28	0.12–0.33	0.14	0.35–0.53	0.12–0.24	0.12–0.16	0.18–0.32	0.15–0.18
**Ovary width**	0.11–0.20	0.18–0.23	0.33–0.36	0.17–0.40	0.21	0.25–0.35	0.12–0.27	0.10–0.11	0.15–0.33	0.15–0.18
**Anterior testis length**	0.16–0.38	0.35	0.44–0.54	0.25–0.83	0.30	0.83–1.17	0.24–0.44	0.22–0.37	0.33–0.83	0.25–0.35
**Anterior testis width**	0.23–0.54	0.26	0.73–0.90	0.32–0.98	0.42	0.51–0.89	0.27–0.56	0.20–0.31	0.33–0.61	0.30–0.38
**Posterior testis length**	0.13–0.41	0.36–0.44	0.44–0.54	0.25–0.90	0.36	0.88–1.22	0.29–0.58	0.28–0.36	0.33–0.83	0.30–0.35
**Posterior testis width**	0.25–0.55	0.34	0.73–0.90	0.32–1.08	0.43	0.60–1.08	0.30–0.67	0.22–0.38	0.40–0.65	0.33–0.43
**Anterior end of vitellarium**	0.31–0.72	–	–	–	–	–	0.43–0.89	0.60	–	–
**Egg length**	0.015–0.030	0.025–0.027	0.029–0.032	0.029–0.032	–	0.025–0.029	0.027–0.031	0.024–0.028	0.026–0.030	0.023–0.028
**Egg width**	0.010–0.018	0.014–0.015	0.015–0.017	0.014–0.017	–	0.013–0.015	0.015–0.019	0.012–0.014	0.136–0.017	0.014–0.016

**Table 3 animals-14-00124-t003:** Genetic *p*-distances (below the diagonal; black) and standard error estimates (above the diagonal; gray) between the species of the family Opisthorchiidae based on partial 28S rRNA gene sequence data.

Opisthorchiidae		1	2	3	4	5	6	7	8	9	10	11	12	13	14	15
**1**	*Metorchis orientalis* MK482051		0.002	0.002	0.003	0.003	0.004	0.004	0.004	0.007	0.006	0.007	0.007	0.007	0.013	0.008
**2**	*Metorchis ussuriensis* KY075777	0.006		0.002	0.003	0.003	0.004	0.004	0.004	0.007	0.006	0.007	0.007	0.007	0.013	0.008
**3**	*Metorchis butoridi* OR095627-OR095634	0.003	0.005		0.003	0.003	0.004	0.004	0.004	0.006	0.006	0.007	0.007	0.007	0.013	0.008
**4**	*Clonorchis sinensis* JF823989	0.009	0.009	0.008		0.002	0.004	0.004	0.003	0.006	0.006	0.006	0.007	0.007	0.013	0.008
**5**	*Opisthorchis felineus* MF099790	0.011	0.011	0.010	0.006		0.003	0.004	0.003	0.006	0.006	0.006	0.007	0.007	0.013	0.008
**6**	*Opisthorchis viverrini* HM004188	0.019	0.019	0.018	0.015	0.015		0.004	0.004	0.007	0.006	0.007	0.007	0.007	0.013	0.008
**7**	*Opisthorchis noverca* KC295443	0.020	0.020	0.018	0.016	0.017	0.017		0.005	0.007	0.006	0.007	0.007	0.007	0.013	0.008
**8**	*Erschoviorchis anuiensis* MK877245	0.015	0.015	0.014	0.014	0.014	0.022	0.025		0.006	0.006	0.006	0.007	0.007	0.013	0.008
**9**	*Amphimerus ovalis* AY116876	0.046	0.048	0.045	0.045	0.045	0.052	0.052	0.042		0.007	0.007	0.008	0.007	0.013	0.008
**10**	*Apophallus zalophi* MG806918	0.043	0.045	0.042	0.042	0.042	0.044	0.050	0.041	0.056		0.005	0.007	0.007	0.012	0.008
**11**	*Liliatrema skrjabini* MT303944	0.048	0.049	0.047	0.046	0.048	0.050	0.052	0.047	0.059	0.028		0.007	0.006	0.013	0.008
**12**	*Cryptocotyle lata* MH025622	0.059	0.061	0.058	0.058	0.060	0.065	0.062	0.060	0.067	0.059	0.054		0.004	0.013	0.008
**13**	*Cryptocotyle lingua* AY222228	0.058	0.060	0.057	0.057	0.059	0.062	0.063	0.055	0.064	0.053	0.047	0.021		0.012	0.008
**14**	*Pachytrema calculus* AF151942	0.209	0.214	0.212	0.209	0.209	0.207	0.215	0.212	0.219	0.212	0.216	0.203	0.210		0.012
**Heterophyidae (outgroup)**																
**15**	*Metagonimus suifunensis* KX387456	0.065	0.063	0.064	0.060	0.064	0.065	0.070	0.065	0.074	0.065	0.063	0.069	0.066	0.198	

**Table 4 animals-14-00124-t004:** Genetic *p*-distances (below the diagonal; black) and standard error estimates (above the diagonal; grey) between the species of the family Opisthorchiidae based on partial *cox*1 mtDNA gene sequence data.

Opisthorchiidae		1	2	3	4	5	6	7	8	9	10	11	12	13	14	15	16	17	18
**1**	*Metorchis butoridi* OR088532		0.001	0.000	0.002	0.002	0.002	0.002	0.003	0.010	0.010	0.010	0.010	0.011	0.011	0.011	0.011	0.012	0.013
**2**	*Metorchis butoridi* OR088533	0.002		0.001	0.001	0.001	0.001	0.001	0.003	0.010	0.010	0.011	0.010	0.011	0.011	0.011	0.011	0.012	0.013
**3**	*Metorchis butoridi* OR088534	0.000	0.002		0.002	0.002	0.002	0.002	0.003	0.010	0.010	0.010	0.010	0.011	0.011	0.011	0.011	0.012	0.013
**4**	*Metorchis butoridi* OR088535	0.004	0.002	0.004		0.001	0.002	0.002	0.003	0.010	0.010	0.011	0.010	0.011	0.011	0.011	0.011	0.012	0.013
**5**	*Metorchis butoridi* OR088536	0.004	0.002	0.004	0.002		0.002	0.002	0.003	0.010	0.010	0.011	0.010	0.011	0.011	0.011	0.011	0.012	0.013
**6**	*Metorchis butoridi* OR088537	0.004	0.002	0.004	0.004	0.004		0.002	0.003	0.010	0.010	0.011	0.010	0.011	0.011	0.011	0.011	0.012	0.013
**7**	*Metorchis butoridi* OR088530	0.003	0.001	0.003	0.003	0.003	0.003		0.003	0.010	0.010	0.011	0.010	0.011	0.011	0.011	0.011	0.012	0.013
**8**	*Metorchis butoridi* OR088531	0.013	0.012	0.013	0.012	0.010	0.013	0.012		0.010	0.010	0.011	0.011	0.011	0.010	0.011	0.011	0.013	0.013
**9**	*Metorchis ussuriensis* OR095177	0.135	0.134	0.135	0.132	0.134	0.132	0.134	0.137		0.010	0.011	0.010	0.011	0.010	0.012	0.011	0.012	0.014
**10**	*Metorchis orientalis* KT239342	0.145	0.145	0.145	0.145	0.145	0.147	0.146	0.153	0.147		0.009	0.011	0.011	0.011	0.011	0.012	0.012	0.013
**11**	*Opisthorchis viverrini* JF739555	0.151	0.151	0.151	0.151	0.151	0.153	0.152	0.158	0.157	0.101		0.010	0.011	0.011	0.011	0.011	0.013	0.013
**12**	*Clonorchis sinensis* MT607652	0.161	0.159	0.161	0.159	0.159	0.161	0.159	0.166	0.148	0.158	0.155		0.010	0.010	0.011	0.012	0.013	0.014
**13**	*Opisthorchis felineus* EU921260	0.159	0.157	0.159	0.155	0.155	0.159	0.157	0.157	0.152	0.156	0.166	0.151		0.010	0.011	0.012	0.013	0.013
**14**	*Opisthorchis sudarikovi* MK033132	0.143	0.141	0.143	0.140	0.140	0.143	0.141	0.145	0.149	0.143	0.140	0.145	0.136		0.011	0.012	0.012	0.014
**15**	*Erschoviorchis anuiensis* OR084097	0.161	0.160	0.161	0.159	0.160	0.162	0.161	0.164	0.178	0.164	0.166	0.158	0.165	0.159		0.011	0.013	0.013
**16**	*Amphimerus* sp. MK238506	0.178	0.179	0.178	0.178	0.179	0.181	0.180	0.183	0.192	0.199	0.183	0.199	0.193	0.191	0.191		0.011	0.013
**17**	*Cryptocotyle lingua* EU876333	0.222	0.224	0.222	0.222	0.223	0.224	0.225	0.227	0.231	0.231	0.224	0.229	0.235	0.225	0.239	0.192		0.013
**Heterophyidae (outgroup)**																			
**18**	*Metagonimus suifunensis* MK736830	0.243	0.243	0.243	0.244	0.244	0.245	0.242	0.247	0.258	0.251	0.244	0.251	0.249	0.244	0.248	0.232	0.240	

## Data Availability

The sequence data analyzed during the current study are available in the National Center for Biotechnology Information database (NCBI, https://www.ncbi.nlm.nih.gov, accessed on 7 July 2023).

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
