# Peer review of "Morpho-Molecular Features and Phylogenetic Relationships of Metorchis butoridi Oschmarin, 1963 (Trematoda: Opisthorchiidae) from East Asia"

_animals, 2023, doi:10.3390/ani14010124_

Round 1
Reviewer 1 Report
Comments and Suggestions for Authors
This is an interesting paper and is a useful contribution, although leaving a lots of questions for later consideration.
Some of the literature on Metorchis butoridi has not been referred to. It was placed in the genus Metametorchis Morozov, 1939 by Filimonova in Sonin (1986). They reported this species in Butorides striata and Phalacrocorax carbo in eastern Siberia and the Russian Far East. Later reports under this name include records from Larus argentatus and Larus mongolicus from the Baikal Region (Nekrasov et al., 1999, Dorzhiev et al. 2020). Scholz (2008) considered Metametorchis a synonym of Parametorchis Skrjabin, 1913. The results of this paper cast considerable doubt on the placing of M. butoridi in this genus. This should be commented on.
Dorzhiev Ts. Z., Badmaeva E. N., Dugarov Zh. N. (2020). Ecological and faunistic analysis of helminths of water-marsh birds of Baikal Siberia: 3. Gulls Laridae. Nature of Inner Asia. â„–1 (14), 66-78. (In Russian). DOI: 10.18101/2542-0623-2020-1-66-78
Nekrasov, A.V., Pronin, N.M., Sanzhieva, S.D. and Timoshenko, T.M. (1999). Diversity of the helminth fauna of the herring gull (Larus argentatus) of Lake Baikal: features of spatial distribution and infestation. Parzazitologiya, 33(5), 426-436. (In Russian)
Sonin, M. D. (1986). Keys to the identification of trematodes of fish-eating birds of the Palaearctic (opisthorchids, renicolids, strigeids). Nauka, Moscow.. 215 pp. (In Russian).
The citation of some papers is perfunctory and should be improved: For example:
The first reference:
Bray, R.A.; Gibson, D.; Jones, A. Keys to the Trematoda, Vol. 3; CAB International and Natural History Museum: London, UK, 2008, 582. doi:10.1079/9780851995885.0000
should be cited as:
Scholz, T. (2008). Family Opisthorchiidae Looss, 1899. In: Bray, R. A.; Gibson, D. I.; Jones, A. (Eds). Keys to the Trematoda. Vol. 3. Wallingford: CAB International and the Natural History Museum, pp. 9–49.
The third reference:
Skrjabin, K.I. Trematodes of man and animals, Osnovy trematodologii; Nauka: Moscow, USSR, 1950.
should be cited as:
Skrjabin, K. I.; Petrov, A. M. (1950). Superfamily Opisthorchoidea Faust, 1929. Osnovy Trematodologii. 4: 81–328. (In Russian).
Comments on the Quality of English LanguageThe English is quite good, and understandable, but is awkward at some points and could be improved by a little editing.
Reviewer 2 Report
Comments and Suggestions for Authors
The topic is important. It is a zoonotic helminth with wide host rage. The problem is least studied, therefore, it will be able to attract a large number of readers. However, the manuscript is not well written. Please follow-
1. Introduction: describe the parasite then pathogenesis and provide knowledge gap.
2. Result: present you data clearly. Here sentences are incomplete.
3. Discussion: Discuss you data on the basis of importance of results.
4. Overall: most problem with the language. It must be thoroughly checked before resubmission.
Comments on the Quality of English LanguageMost problem with the language. Many sentences are incomplete. It must be thoroughly checked before resubmission.
Reviewer 3 Report
Comments and Suggestions for Authors
The manuscript by Solodovnik and his colleagues is devoted to the species diagnosis of Metorchis trematodes experimentally obtained from metacercariae parasitising freshwater fish in Russian Far East. To achieve their goals, the authors used both morphological and molecular phylogenetic methods. It is pleasant to note that the authors are one of the few who submit articles to MDPS journals that correctly use the Latin names of organisms in accordance with the rules of the International Code of Zoological Nomenclature. I congratulate the authors for the extensive effort made in carrying out a high-quality scientific paper. I read the article with great interest. The work of Russian colleagues undoubtedly meets the goals and objectives of the journal Animals and can be published.
However, I have a few comments on the manuscript, which will undoubtedly improve the article.
1. I found the Simple Summary to be too short and too simple. It is necessary to add information about the parasite (trematode) and its hosts it parasitizes.
2. Molecular phylogenetic analysis was carried out competently, but, Please, remove supports in polytomic nodes (In results, section 3.3.2). It brings sense in dichotomic ones only. You can replace the trees with "contype=allcompat" option in MrBayes. But simple removing of the support levels is way cheaper.
Line 72 – Please, use more common term – “whole mounts”.
Lines 113-126 – Please, remove this information here. It is already contained in the more suitable section – Institutional Review Board Statement.
Line 207 – correct term “Intraspecific Variability”. The term “Intraspecific Variation” more suitable for plant growth.
Line 297 - A sentence cannot begin with an abbreviation. Therefore, in such cases, the generic name is given in full – “Metorchis ussuriensis differed …” Please check all text.
It is necessary to formalize the Author Contributions in accordance with the rules of MDPI journals.
The manuscript may be published in Animals, but small corrections are needed.
Round 2
Reviewer 2 Report
Comments and Suggestions for Authors
Data sets will definitely contribute the understanding of the genus metorchis, however, still presentation style had not been improved. I thought in the revised version, it will be improved and then I will go for the detail reviewing. The following points must be addressed-
1. Title: Too long. Revise as " Morpho-molecular features, and phylogenetic relationships of Metorchis butoridi Oschmarin, 1963 (Trematoda: opisthorchiidae) from East Asia
Abstract:
Line 14: delete individual of the
Line 17-18: delete form ' these data ......important species'
Line 22: delete s from 'opisthorchiids'
Line 24: insert 'were confirmed as' instead of ' proved to belong to the,
Line 25: delete 'a set of' species'
Introduction
Highlight zoonotic impacts (pls see Anisuzzaman et al 2023)
Highlight fish species as intermediate hosts. Please See Labony et al 2020 among others
While highlighting Southeast Asian Species 'Metorchis Orientalis' then read previous literatures on avian liverflukes, including 'Avian liver fluke infection in indigenous ducks in Bangladesh: prevalence and pathology"
Materials and Methods
Line 84-86: partially true, since metacecariae of Metorchis can easily be separated from others. Please see Labony et al 2020. Thus, modify the statement.
Results
Major problem is in the section 3.2. This section has not been written properly. Most of the sentences are in complete, without verbs. You must this section following grammatical rules and in past tense. Please be careful during the revised version.
Fig 2: Please indicate the reference sequence (Metorchis butoridi) and sequences of the present study in the phylogram and compare with the reference sequence to claim the new isolate as Metorchis butoridi, otherwise it seems to be M. orientalis.
Fig 3. Please put one sequence of Metorchis butoridi as a reference sequence to claim the new isolate as Metorchis butoridi, otherwise it seems to be M. orientalis.
Fig 5: Same here as well.
Discussion
Delete subheadings
Conclusions
Another big problem. Avoid putting reference
Please draw conclusions directly from your results
References
Update
Comments on the Quality of English Language
Major editing
